# Autologous Stem Cell Transplant in Hodgkin’s and Non-Hodgkin’s Lymphoma, Multiple Myeloma, and AL Amyloidosis

**DOI:** 10.3390/cells12242855

**Published:** 2023-12-18

**Authors:** Sulaiman Mohammed Alnasser, Khalid Saad Alharbi, Ali F. Almutairy, Sulaiman Mohammed Almutairi, Abdulmalik Mohammed Alolayan

**Affiliations:** 1Department of Pharmacology and Toxicology, Unaizah College of Pharmacy, Qassim University, Buraydah 51452, Saudi Arabia; khalid.alharbi9@qu.edu.sa (K.S.A.); a.almotairi@qu.edu.sa (A.F.A.); 2Unaizah College of Pharmacy, Qassim University, Buraydah 52571, Saudi Arabia; sulaiman.m.almtairi@gmail.com (S.M.A.); ph.abdulmalikolayan@gmail.com (A.M.A.)

**Keywords:** autologous stem cell transplantation, cancers, oncology, Hodgkin’s lymphoma, non-Hodgkin’s lymphoma, multiple myeloma, AL amyloidosis

## Abstract

Human body cells are stem cell (SC) derivatives originating from bone marrow. Their special characteristics include their capacity to support the formation and self-repair of the cells. Cancer cells multiply uncontrollably and invade healthy tissues, making stem cell transplants a viable option for cancer patients undergoing high-dose chemotherapy (HDC). When chemotherapy is used at very high doses to eradicate all cancer cells from aggressive tumors, blood-forming cells and leukocytes are either completely or partially destroyed. Autologous stem cell transplantation (ASCT) is necessary for patients in those circumstances. The patients who undergo autologous transplants receive their own stem cells (SCs). The transplanted stem cells first come into contact with the bone marrow and then undergo engraftment, before differentiating into blood cells. ASCT is one of the most significant and innovative strategies for treating diseases. Here we focus on the treatment of Hodgkin’s lymphoma, non-Hodgkin’s lymphoma, multiple myeloma, and AL amyloidosis, using ASCT. This review provides a comprehensive picture of the effectiveness and the safety of ASCT as a therapeutic approach for these diseases, based on the currently available evidence.

## 1. Understanding Autologous Stem Cell Transplants

### 1.1. Stem Cells (SCs)

SCs are primary human body cells, with the capability to divide indefinitely and develop into specialized cells for the body tissues, as shown in Figure 1. These cells have a critical role in the repair of lost or damaged cells because of illness or injury. SCs have also drawn a lot of interest as a potential therapeutic approach for a diverse range of diseases and conditions [1].

SCs have the capacity to develop into over 200 different type of body cells. It was once thought that SCs could only differentiate into adult cells of the same organ. However, it is now known that they can develop into different types of cells, including ectoderm, mesoderm, and endoderm cells. Moreover, various tissues, such as bone marrow, the heart, the kidney, the liver, and others, have different numbers of SCs [2].

### 1.2. SC Classification Based on Source and Potency

SCs are categorized according to their potential and the location in where they are obtained or harvested [3]. They can be separated depending on how potent they are, or how well they can differentiate into different cell types. Hence, some researchers divide them into four primary categories: totipotent, pluripotent, multipotent, and unipotent [4], while others extend the classification to five by including oligopotent as well. All categories are described in Table 1 [5]. 

On the other hand, SCs can be categorized based on their sources, distinguishing them into two groups: early or embryonic and mature or adult (as shown in Table 2) [11]. After around five days of development, early SCs, also known as ESCs, are identified in the internal cell mass of a blastocyst. Moreover, mature SCs are recognized in specific adult body tissues, such as the placenta and umbilical cord blood (UCB), following delivery [12]. Additionally, the primordial SCs (or pluripotent SCs), are capable of producing every type of cell in any organism [13].

### 1.3. SC Transplant

SC transplant can be employed to treat various cancers by replacing blood-forming SCs in individuals with cells damaged by high chemotherapy doses and radiation. Blood-forming SCs used in transplants can be sourced from the bone marrow, bloodstream, and UCB [20]. 

SC transplants have also been employed in the treatment of various disorders, including autoimmune [21], neurological [22], and genetic conditions [23,24]. Moreover, Shah et al. (2016) highlighted the use of SCs as a therapeutic option for aggressive disorders, such as brain tumors [25]. This approach is considered innovative as it specifically targets cancer SCs while overcoming the limitations of traditional cancer therapy methods, such as surgery [20]. Previously, this therapy was only utilized as an adjunctive step in traditional approaches [26]. According to Marofi et al. (2021), SC therapy has demonstrated effectiveness in treating diseases, such as leukemia and multiple myeloma, in various scenarios [27].

There are two primary categories of transplants, named based on the source of SCs (autologous or allogeneic). Autologous refers to a medical treatment or procedure that uses tissues, cells, or blood products that originate from the same individual. In contrast, “allo” means other; thus, in allogenic transplants, the patients receive SCs from healthy donors (ASCO) [28]. 

## 2. The Role of ASCT in Cancer Management

The main application of ASCT is the management of cancer. The effectiveness of myelosuppressive chemotherapy or radiation in treating various cancers depends on the dosage. An anticancer response can be significantly enhanced at higher doses. This leads to bone marrow suppression, causing dose-limiting toxicity. Additionally, the infusion of autologous HSCT is carried out after high-dose chemotherapy (HDC) or radiation, forming a “preparative regimen” to restore blood production and immunity. This approach has proven effective against various solid tumors and specific hematological malignancies [29,30]. 

HSCT has become a well-established life-saving therapeutic method for numerous patients with hematological malignancies, innate defects, or bone marrow failure syndromes. However, the field of SC transplants and cellular therapies is evolving rapidly, with changing transplant practices and an increasing number of long-term survivors. Consequently, there is a growing need for ongoing education among healthcare professionals, including doctors, nurses, and other practitioners involved in SC transplants and cellular therapies [31].

### 2.1. The Role of ASCT in Treating Hodgkin’s Lymphoma (HL)

The third most common tumor in the head and cervical region is lymphoma, arising from the lymphoreticular system. Moreover, HL and non-Hodgkin’s lymphoma (NHL) constitute two main types of lymphomas in this context [32]. A rare B-cell malignant tumor arising from aberrant B-cell clonal augmentation is known as HL and constitutes approximately 15% of all lymphomas [33]. In 2020, HL accounted for 0.4% of all recently reported cancer-related incidents and 0.2% of mortality worldwide [34]. Despite being relatively rare, HL stands out as the most prevalent cancer among adolescents aged 15 to 19 years [35].

However, it is categorized as either classical HL (cHL) or nodular lymphocyte predominant HL. In addition, there are four distinct subcategories under cHL, including nodular sclerosis, lymphocyte depletion, mixed cellularity, and lymphocyte-rich HL. Moreover, the phase of the disease affects the selection of the most effective therapy. Positron emission tomography and prognostic models determine if the patients are at a higher or lower risk of disease recurrence. Hence, they are used to choose the most effective treatment option and to guide treatment decisions [36].

The initial treatment for HL patients is determined by the histology and anatomical phase of the disease, alongside negative predictive indicators, using combined modality regimens, which involve brief sessions of chemotherapy accompanied by involved-field radiation therapies (IFRT). On the other hand, patients diagnosed with advanced-stage disease are given a lengthier session of chemotherapy, frequently without radiotherapy [37]. However, antibodies, such as anti-programmed death-1 (PD-1) and the antibody–drug conjugate brentuximab vedotin (BV), are often utilized in frontline therapies. However, for the majority of patients who experience recurrence after initial therapies, HDC followed by an ASCT is the recommended method of treatment [38]. Moreover, there are two approaches in which ASCT acts in HL, namely single ASCT and tandem ASCT, each with a distinct role.

### 2.2. The Role of Single ASCT in the Treatment of HL

The effectiveness of ASCT in HL has been reviewed previously; the first study, carried out by Schmitz et al. (2002) focused on HL patients with chemosensitivity after their initial recurrence. The study observed that 55% of patients randomly assigned to ASCT achieved progression-free survival (PFS) after 3 years, compared to 34% of those who underwent intensive conventional chemotherapy (CHT) alone. However, the study did not show a significant change in overall survival (OS) [39]. The second study aimed to investigate whether HDC followed by ASCT should be employed as early consolidative therapy in patients with advanced, unfavorable HL. The results indicated that the role of HDC followed by ASCT as a frontline therapy in people with advanced HL was without benefit [40]. The results of the extended follow-up of the work by Federico et al. (2003) showed that the 10-year OS was 85% and 84% for patients undergoing HDC followed by ASCT or CHT alone, respectively, with no statistically significant differences (*p* = 0.7). Contrarily, for 10 years, relapse-free survival (RFS) and failure-free survival (FFS) were 88% and 79%, respectively, for patients undergoing HDC followed by ASCT, compared to those who received CHT alone at 89% and 75% (*p* = 0.7–0.8), respectively. The researchers concluded that HDC followed by ASCT consolidation is inferior to initial CHT. Hence, patients responsive to their initial CHT should not be offered consolidation therapy [41,42].

Cortez et al. (2011) conducted a retrospective study utilizing data from patient charts to evaluate 106 patients with cHL who experienced recurrence or were refractory to treatment. These patients had undergone HDC followed by ASCT at a single transplant facility between April 1993 and December 2006. The study revealed that at five and ten years, the OS for this population was 86% and 70%, respectively. Additionally, the five-year PFS was approximately 60%, with four deaths attributed to medical procedures. Recurrence of cHL following the transplant was the most common cause of mortality. Univariate analysis indicated that hemoglobin levels below 10 g/dL at diagnosis and responsiveness to pre-transplant therapy influenced patient survival. The findings suggest that ASCT is a successful treatment for pre-transplant chemotherapy-responsive cases of both early and late recurrence in cHL. Furthermore, post-transplant patients’ survival is negatively affected by hemoglobin levels below 10 g/dL at the time of HL diagnosis [43].

Castagna et al. (2015) reviewed the current role of ASCT for recurrence or refractory cHL. He concluded that using HDC followed by ASCT can benefit 30–80% of patients with recurrent cHL one year after transplantation [39,44,45]. However, 50% of ASCT patients experience either a recurrence or progression of the disease. For this reason, the randomized studies failed to report a significant improvement in OS, which is likely due to the “cross over” to ASCT of patients who did not respond to conventional therapy [46].

Auto-HSCT remains the recommended course of treatment for patients with recurrent or refractory cHL [47]. Two landmark randomized clinical trials (RCTs) were carried out in 1993 and in 2002, which aimed to compare HDC followed by auto-HSCT versus CHT alone. The results demonstrated a great benefit for auto-HSCT over CHT in event-free survival (EFS) and freedom from treatment failure (FFTF), but no significant advantage regarding OS [39,44]. Nevertheless, according to Sureda et al. (2015) and the European Society for Blood and Marrow Transplantation (EBMT), recent recommendations for auto-HSCT in cHL suggest that it is the standard of therapy in cases of sensitive recurrence or second complete response. However, it is generally not advised in cases of first complete remission. In the case of refractory disease, it remains a clinical option [48]. 

### 2.3. The Role of Tandem ASCT for Treating HL

To enhance the outcomes after the transplant of patients having low risk factors, a research group has investigated a tandem transplants method. The study demonstrated the practicability of tandem auto-HSCT and the related non-relapse mortality (NRM) of 0–5%, OS of 54–84%, and PFS of 49–55% at 5-years [49]. Moreover, currently, cHL that has recurred after undergoing auto-HSCT is approved for therapy with the drug BV [50,51]. Based on these findings, risk-adapted tandem auto-HSCT can be considered as a treatment choice for low-risk patients. However, the incorporation of positron emission tomography outcomes and new medications, like BV and checkpoint inhibitors, could aid in narrowing the necessity for a second auto-HSCT and potentially improving patient outcomes [48].

The role of ASCT in treating HL is being investigated, but there is still no conclusive evidence that it improves OS, despite several studies demonstrating favorable results in terms of PFS. However, recent studies have shown that pre-transplant chemotherapy-responsive cases of both early and late recurrence in cHL can be successfully treated with ASCT. BV may reduce the requirement for another auto-HSCT and also enhances patient outcomes.

### 2.4. The Role of ASCT in Treating NHL

NHL is a set of lymphoproliferative cancers that often originates in lymphoid tissue. It constitutes approximately 5% of head and neck cancers and manifests with a range of symptoms similar to those of Hodgkin’s disease [32]. Both HL and NHL can occur in the neck and head region. However, NHL patients are more likely to develop extra-nodal diseases, regardless of whether lymph nodes are involved [52]. Various categories are used to classify NHL, with the WHO classification for lymphoma malignancies being the most frequently followed [32]. NHL is one of the most prevalent hematological malignancies, accounting for approximately 3% of all cancer diagnoses and mortality worldwide [53]. The role of ASCT in various NHLs is highlighted below.

#### 2.4.1. ASCT in Peripheral T-Cell Lymphoma (PTCL)

PTCL is a diverse subset of lymphomas that arise from the T-cell lineage, and they make up between 10% and 15% of the entire NHL [54]. Zahid et al. (2017) reviewed the role of ASCT in T-cell lymphoma (TCL). However, due to lack of adequately powered RCTs, the role of ASCT in TCL is little understood [42]. Moreover, D’Amore et al. (2012), directed a prospective phase II study aimed at measuring the effectiveness of a dose-dense strategy combined with upfront HDT, followed by ASCT in systemic PTCL. Nevertheless, CHOEP-14 (CHOP + etoposide) or CHOP-14 for adults over 60 was used to treat patients with systemic PTCL, showing a 51% 5-year OS [55]. 

In a study by Beitinjaneh et al. (2015), 76 patients underwent transplant after their initial recurrence, and T-cell lymphoma (TCL) patients underwent ASCT in the frontline setting. ASCT patients who received BEAM (carmustine + etoposide + cytarabine + melphalan) showed better 4-year OS and PFS in the first complete response. Additionally, patients with diseases sensitive to chemotherapy who achieved a CR with ASCT demonstrated an 84% 4-year OS compared to 44% for those with partial responses [56]. However, poor outcomes in some PTCL subgroups following CHT have led to a trend of consolidating HDC followed by ASCT [57,58,59,60]. In a study carried out by Reimer et al. (2009), patients having PTCL who underwent ASCT as first-line therapies had an assessed 3-year OS of 48% and PFS of 36%. Patients having HDC followed by ASCT had an estimated 71% 3-years OS, compared to 11% for those who did not [57]. Owing to the absence of sufficiently powered RCTs, the role of ASCT in the PTCL is not fully understood, and more studies are required. However, several studies have shown that ASCT can improve OS and PFS in some PTCL subgroups, particularly in patients with chemosensitive diseases who achieve remission. 

#### 2.4.2. The Role of ASCT in B-Cell NHL

It is anticipated that B-cell NHL made up 86% of all lymphoid malignancies identified in 2016, and the most prevalent subtype of lymphoma is diffuse large B-cell lymphoma (DLBCL) [61]. However, Stiff et al. (2013) studied the role of HDC followed by ASCT in RCT phase III as a consolidation therapy for patients with aggressive forms of NHL. The results indicated no benefits in OS, although they did observe an improvement in PFS [62]. Tarella et al. (2007), conducted a study on 112 patients with DLBCL who had undergone HDC followed by ASCT. The results demonstrated that over 80% of the patients achieved remission, with a 4-year OS and event-free survival (EFS) estimated at 76% and 73%, respectively [63]. Furthermore, a phase II study conducted by Vitolo et al. (2009) aimed to compare the addition of rituximab to HDC followed by ASCT with those who did not add rituximab in untreated patients with international prognostic index (IPI) high-intermediate/high-risk DLBCL. The result indicated that the 4-year OS was 80% and 54%, respectively [64]. Moreover, there was a retrospective study published by Kim et al. (2016), that aimed to evaluate the impact of upfront ASCT in patients with various molecular subtypes of advanced-stage DLBCL (germinal center B-cell-like (GCB)) and the non-GCB subtype. According to the findings, the ASCT group benefited considerably more from OS and PFS than those in the non-ASCT group. However, patients in the non-ASCT group with the non-GCB subtype tended to have poor outcomes. In addition, in the ASCT group, the two subtypes did not significantly differ from one another [65]. Kaneko et al. (2015) recommended that upfront ASCT consolidation might be preferable for treating a subset of high-risk non-GCB lymphomas. However, for high-risk DLBCL in CR1, upfront HDC followed by ASCT might result in better outcomes [66].

Wullenkord et al. (2021) carried out a retrospective study with information from a single-center analysis, which aimed to analyze the features and outcomes of patients treated through HDC followed by ASCT in various types of DLBCL (from 2002 to 2019). A total of 247 DLBCL patients received HDC followed by ASCT, either as consolidation after first-line therapies or just after salvage therapies for recurrent conditions. The results showed that after 3 years, the PFS was 63%, and OS was 68%, with 28% of the patients experiencing a recurrence following ASCT. Multivariate analysis revealed that remissions at day 100 post-ASCT, age at ASCT, and the quantity of injected CD34+ cells were independent prognostic variables for both PFS and OS. Initial recurrence within 12 months following first-line therapies was associated with worse PFS and OS for patients with DLBCL and HL. The leading causes of mortality post-ASCT were infections (23%) and lymphoma recurrences and/or progressions (64%). The authors concluded that, in the era of novel targeted medicines, HDC followed by ASCT remains an effective and viable therapy for high-risk or recurrent DLBCL, with prognostic significance linked to the degree of remission, age at ASCT, and the quantity of infused SCs [67].

Schmitz et al. (2018) carried out a review about auto-HSCT in DLBCL. However, auto-HSCT continues to be considered the preferred therapy for refractory or recurrent DLBCL patients. Nevertheless, auto-HSCT has minimal benefits even for patients who attain either PR or CR after salvage therapies and resiniferatoxin (RTX), and the outcomes are not as promising as they were earlier [68,69]. Moreover, the prolonged remissions following auto-HSCT are uncommon in patients with refractory illness or early recurrence pre-treated using RTX as part of induction therapy [70]. According to Sureda et al. (2015), it is the standard therapy in cases of sensitive recurrence or second complete response. However, in cases of refractory disease and first complete remission, it is considered a clinical option. The guidelines emphasize that auto-HSCT is not recommended when a patient’s condition is refractory and not improving with salvage therapy [71]. In general, auto-HSCT is not recommended as a first-line therapy for DLBCL, despite promising new evidence on PET guidance [72]. Hence, the authors do not recommend auto-HSCT in individuals with refractory illnesses who do not respond to salvage therapies [70].

In summary, auto-HSCT proves to be the most effective therapeutic approach, contingent on factors, such as remission status, age, and the quantity of infused stem cells. HDC followed by ASCT demonstrates potential in enhancing OS and PFS in DLBCL patients; however, additional research is necessary to delineate its role in aggressive NHL cases.

#### 2.4.3. Role of the ASCT in Follicular Lymphoma

A review conducted by Robinson (2018) focused on ASCT in follicular lymphoma (FL) as a first-line response [73]. The utilization of ASCT gained attention in the 1980s, challenging the inefficacy of SDT for indolent lymphoma treatment. Preliminary studies showing promise prompted the design of significant randomized trials comparing ASCT to either no additional therapy or interferon. Despite multiple studies, including those conducted by Lenz et al. (2004) and Ladetto et al. (2008), revealing enhanced disease control, there is no demonstrated OS benefit [74,75]. Furthermore, the use of ASCT as a first-line consolidation approach has been discontinued due to these findings and a better understanding of the acute and prolonged toxicity associated with ASCT. However, the CUP trial, a randomized study conducted by Schouten et al. (2003), compared consolidation with ASCT (using stem cells that underwent purging or not) without further therapy in recurrent patients. In this trial, 140 follicular lymphoma patients with recurrences were randomly assigned to receive either CHT or an auto-HSCT for consolidation. As a result, patients who received CHT alone had a 2-year PFS of 26%, compared to 58% for patients who underwent HSCT. Additionally, the two transplanted arms showed an OS advantage [76]. The authors concluded that auto-HSCT should not be used in the first response; however, when a patient’s condition has recurred and is responding to reinduction therapy, auto-HSCT should be considered [73].

In conclusion, due to the absence of proven OS benefits and a greater understanding of both short-term and long-term toxicity associated with ASCT, the use of ASCT as a first-line consolidation approach for FL has been discontinued. However, the CUP study has demonstrated that ASCT can enhance PFS and OS in recurrent FL cases responding to reinduction therapies, making it a viable treatment option that should be considered.

#### 2.4.4. Role of the ASCT in Treating Multiple Myeloma (MM)

MM is a hematological malignancy characterized by the presence of abnormally clonal plasma cells within the bone marrow [77]. It constitutes 10% of all hematological malignancies and 1% of all malignancies overall [78]. The gold standard for treating healthy, young patients with MM, whether newly diagnosed or experiencing a recurrence, is auto-HSCT. Despite the expansion of therapeutic options for MM treatment over the past decade, recent studies have indicated that early auto-HSCT transplantation is preferable to a later stage (during recurrence) [79,80]. 

The number of patients undergoing autologous stem cell transplant (ASCT) in older age groups has significantly increased in recent years. While many trials historically used an age threshold of 65 years to select suitable patients for ASCT in MM, numerous studies have shown that age alone has minimal impact on the outcomes of ASCT in MM patients. For instance, an analysis by the Center for International Blood and Marrow Transplant Research (CIBMTR) involving 946 MM patients aged 70 or older at ASCT revealed that those older than 70 who underwent ASCT experienced similar antimyeloma benefits without increased non-relapse mortality, relapse rate, or progression-free survival. Another investigation involving 207 MM patients aged 70–76 who underwent ASCT at the Mayo Clinic demonstrated that ASCT was well-tolerated in this age group and exhibited non-inferior progression-free survival and overall survival compared to patients under the age of 70 [81]. Moreover, transplant suitability is determined in part by assessing severe comorbidities and fragility, both of which need cautious patient selection. Patients with certain conditions, such as severe heart failure or poor performance status, are generally not recommended for ASCT. However, it is important to note that renal impairment does not necessarily preclude the use of ASCT [82]. Additionally, a comprehensive database study indicated that patients with moderate to severe renal impairment at the time of ASCT demonstrated good tolerance for the procedure [83]. This suggests that, in some cases, renal impairment may not be a strict contraindication for ASCT, and the decision should be individualized based on the patient’s overall health and specific medical conditions. There are two approaches in which ASCT acts in MM, namely single ASCT or tandem ASCT, and each has a discrete role.

#### 2.4.5. The Role of Single ASCT in Treating MM

The goal of induction chemotherapy before HDC and ASCT is to reduce the tumor burden while minimizing toxicity to healthy hematopoietic cells, thereby enhancing response rates and increasing the likelihood of successful engraftment. In this context, alkylating drugs were often avoided in induction regimens, and dexamethasone-based regimens, such as VAD (vincristine, doxorubicin, and dexamethasone), were used before the introduction of novel agents. This approach aimed to optimize the conditions for a successful transplant and improve overall treatment outcomes [84]. Since novel drugs have been available, several studies have demonstrated that induction regimens with one or two novel drugs (thalidomide or bortezomib) are superior to VAD regimens in terms of elevating rates of CR, CR plus near-complete response (nCR), or VGPR both prior to and following ASCT [85,86,87,88].

Several trials have demonstrated the efficacy of combining bortezomib, dexamethasone, and thalidomide (VTD). In the GIMEMA-MMY-3006 phase III trial, patients who underwent VTD vs. TD as induction (three cycles) and consolidation (two cycles) following two phases of ASCT demonstrated superior responses compared to those who underwent TD alone (≥PR: 93% vs. 79%, *p* < 0.0001; ≥VGPR: 62% vs. 28%, *p* < 0.0001), which translated into a significantly improved PFS (HR = 0.62, *p* < 0.0001) [89]. In addition, even though the IFM2013-04 study did not identify an important variation in CR rates, it demonstrated the superiority of the VTD arm in terms of ORR (92.3% vs. 83.4%), as opposed to the bortezomib, cyclophosphamide, and dexamethasone (VCD) arms [90]. However, the triplet VTD did not respond better when cyclophosphamide was added; following induction, 51% and 44% of patients in the VTD and VTDC arms, respectively, achieved nCR/CR [91]. Due to its proven value in treating high-risk patients, proteasome inhibitors, like bortezomib, have become indispensable in the context of VTD, which has become the standard induction regimen [92,93].

Another phase III trial by PETHEMA/GEM, contrasting VTD vs. TD vs. VBMCP/VBAD/B, proved the advantages of VTD over TD. It demonstrated enhanced results in terms of CR rates (35% vs. 14%, *p* = 0.001) and median PFS (56.2 months vs. 28.2 months, *p* = 0.01) among those receiving VTD vs. TD for six cycles as induction therapy prior to ASCT. Moreover, using 3–5 cycles of VTD prior to transplant is a standard procedure, while using 6 cycles of VTD was allied with deeper responses. When delivering 6 cycles instead of 3–4, this must be balanced against the increased adverse effects, particularly neuropathy [94].

Likewise, the combination of lenalidomide, bortezomib, and dexamethasone [RVD] with bortezomib increased the OS (75 vs. 64 months; *p* = 0.025) and median PFS (43 vs. 30 months; *p* = 0.002) considerably when compared to Rd alone. Consequently, RVD is now considered to be the standard therapy for those who have recently been diagnosed with MM, leading the IFM to introduce VRD as induction therapy [94,95,96].

These outcomes have led to the standardization of treatment for induction therapy in recently diagnosed MM patients who are transplant candidates to be VTD, particularly in Europe [97].

Lenalidomide was initially contrasted with Rd in the phase III SWOG S0777 study, which included participants of various ages and without the intention of early ASCT. This was carried out in place of thalidomide when combined with bortezomib and dexamethasone (VRD). The study’s primary objective, median PFS, was found to be 41 months for VRD and 29 months for Rd following a median follow-up of 84 months (*p* = 0.003); OS was not met in the first arm and was attained at 69 months in the second arm [98]. The phase III IFM 2009 trial applied three cycles of VRD as induction. On the other hand, following six cycles of VRD, subjects in the PETHEMA/GEM2012 study underwent ASCT conditioning with busulfan + melphalan, as opposed to melphalan. After induction, using next-generation flow (NGF) at a level of 3 × 10^−6^, 83.4% of patients obtained at least PR, 66.6% at least VGPR, 33.4% CR, and 28.8% of patients achieved measurable residual disease (MRD) negativity [99].

Mostly utilized in the United States, the VRD regimen has evolved into another accepted standard of therapy for patients with recently diagnosed TE MM. In a study comprising the largest patient group treated for VRD in the United States (1000 patients, 751 of whom had upfront ASCT), ORR was 97%, ≥VGPR 67.6%, and CR was 35.9% following four induction cycles. Patients who underwent ASCT had a median PFS and OS of 63 months and 123.4 months, respectively, with a median follow-up of 102 months [100]. Similar results were observed in a different “real-life” experience where patients had a median PFS of 50 months and a median OS of 101.7 months after receiving 4–6 VRD cycles with ASCT. A thorough retrospective analysis of randomized trials found that with the same safety profile, there was a much greater VGPR rate following six cycles of VRD (70% vs. 60%, respectively) compared to six cycles of VTD. However, no prospective randomized trial has contrasted VRD with VTD [101,102]. Furthermore, the phase III ENDURANCE study, which included patients without intending to undergo ASCT, did not show that carfilzomib, lenalidomide, and dexamethasone (KRD) was superior to VRD [103].

Moreover, in the FORTE phase II trial, various consolidation and induction therapies were examined (with carfilzomib either alongside or without ASCT). Patients who were randomly allocated to be given four cycles of KRD as induction exhibited considerably greater VGPR, the main outcome, in comparison to those who received four cycles of carfilzomib, cyclophosphamide, and dexamethasone (KCD) (70% vs. 53%, OR = 2.14, *p* = 0.0002) [104]. On the other hand, the advent of monoclonal antibodies (mAbs), like daratumumab, have once again altered the course of MM therapy. Several studies evaluated the effects of mAb inclusion with the aforementioned triple therapy. In the phase III CASSIOPEIA study, the quadruplet was observed to generate at least VGPR in 65% of patients vs. 56% among those receiving VTD. The study compared four cycles of VTD vs. VTS + daratumumab (D-VTD) as induction prior to ASCT [105]. However, D-VTD and VRD were not contrasted directly in any randomized experiment. The phase II GRIFFIN study found that introducing daratumumab to the VRD combination (D-VRD) produced superior responses following four cycles of D-VRD. The VGPR was 72% and MRD negativity was 21.2%, as opposed to 56.7% and 5.8%, respectively, for VRD [106]. In addition, both the safety and effectiveness of D-VRD vs. VRD were compared in the phase III PERSEUS study. In the phase II MASTER trial, 90% of patients obtained VGPR following four cycles of daratumumab and KRD, with 38% showing MRD-negative at a level of 10^−5^ and 24% at a level of 10^−6^ [107]. The non-randomized MANHATTAN trial revealed a 100% ORR after eight sessions of D-KRD (weekly administration of carfilzomib), with 71% patients being MRD-negative and 95% of them achieving at least VGPR. SCs have been made available to patients for ASCT following 4–6 cycles of D-KRD [108].

In the IFM 2018-01 study, at least VGPR was reached in 78% of patients after six cycles. This was by utilizing oral ixazomib in conjunction with lenalidomide, dexamethasone, and daratumumab (IRD-Dara) as induction prior to ASCT, with an MRD negativity of 28% at a level of 10^−5^. The daratumumab with VCD (Dara-VCD) vs. VTD combinations given as induction and consolidation following ASCT were evaluated for effectiveness in the phase II EMN 18 investigation [109]. The triplet and, more significantly, quadruplet combinations, including mAbs, produced superior responses among patients who receive ASCT during induction therapy, with the majority of patients reaching MRD negativity at the level of 10^−5^. The question of whether prolonged induction treatment exceeding 4–6 cycles may result in even greater responses and eliminate ASCT entirely still is yet to be answered.

On the other hand, consolidation therapy has been effective in enhancing the response obtained by the use of ASCT in various recent trials. Following a median follow-up of 124 months in the phase III GIMEMA-MMY-3006 trial, patients being given VTD as induction and consolidation following two rounds of ASCT had a median PFS of 60 months, as opposed to 41 months for those given TD (*p* < 0.0001). At 10 years, the OS was 46% and 60%, respectively (HR = 0.68, *p* = 0.0068). Particularly, CR was raised from 49% to 61% after two cycles of VTD consolidation. This rise was not as significant for TD (from 40% to 47%) [110]. However, obtaining CR following ASCT has been associated with extended OS and PFS [111].

More sensitive methods for response assessment (like MRD) utilized after therapy have shown better results contrasted with conventional CR, developed as a result of the greater frequency and strength of response achieved via novel regimens [112]. Furthermore, whether MRD was detected by next-generation sequencing (NGS) or NGF, negative-MRD status has been linked with a survival advantage. A minimum sensitivity of 10^−5^ is necessary and technique sensitivity limits can affect both PFS and OS. For PFS benefits, the HRs are 0.31 at a level of 10^−5^ and 0.22 at a level of 10^−6^ [113].

Additionally, insignificant variation has been reported in the 3-year PFS among patients with standard vs. high-risk cytogenetics in the PETHEMA/GEM2012 study. Patients who were MRD-negative (45%) with a median detection limit of 3 × 10^−6^, following two VRD cycles as consolidation had an 87% PFS in comparison to 50% for those with persistent MRD (HR 0.21, *p* < 0.001) [114].

Multiple research studies have shown outstanding results when triplet KRD is utilized as consolidation. The sCR following eight cycles of KRD (four as induction treatment and four as consolidation post-ASCT) was 60% in the phase II trial from the Multiple Myeloma Research Consortium (MMRC), which included 76 patients. Of these patients, 52% obtained MRD negativity. The median PFS and OS at five years, following a median follow-up of 56 months, were not achieved in patients who did not achieve MRD negativity. The parameters were 85% for PFS and 91% for OS [115]. Similarly, a phase II trial performed by IFM showed an MRD negativity of 92.6% and 63% at 2.5 × 10^−5^ and 10^−6^ levels, respectively. This was alongside an sCR rate of 62% following consolidation with four KRD cycles. Five-year PFS was 45.1% in all groups, with a median follow-up similar to the MMRC trial (60.5 months), with approximately 60% MRD-negative patients and 35% MRD-positive patients [116].

In the CASSIOPEIA and GRYPHON investigations, quadruplet regimens (containing daratumumab, such as D-VTD and D-VRD (two cycles)) delivered following ASCT, have shown enhanced responses. D-VTd substantially outperformed VTD induction/consolidation in terms of MRD-negative rate (33.7% vs. 19.9%, *p* < 0.0001) and ≥one-year sustained MRD negativity rate (50.1% vs. 30.1%, *p* < 0.0001) in patients who achieved at least CR following consolidation [117]. In the GRYPHON study, MRD negativity for at least 1 year was considerably greater in the D-VRD arm than in the VRD arm (44.2% vs. 12.6%, *p* < 0.0001), depicting a 3-year PFS of 88.9% (vs. 81.2% for VRD). MRD negativity at level of 10^−5^ went up from 22% following induction to 50% after consolidation [118]. The impact of consolidation therapy in the application of ASCT strategy has not been extensively deliberated in prospective studies. With a total of 1503 patients, the EMN02/HOVON95 study contrasted 2 cycles of VRD vs. no consolidation following escalation with either ASCT (single or tandem) or bortezomib, melphalan, or prednisone (VMP). Consolidation was linked to a substantially longer PFS (median 59.3 months vs. 42.9 months, HR = 0.81, *p* = 0.016) following a median follow-up of 74.8 months. The OS was not reached in either arm, despite the OS curve showing a favorable trend for consolidation after 5.6 years [119].

Remarkably, following consolidation, the median PFS for patients with MRD-negative was 87 months in comparison with 38 months for MRD-positive patients (HR = 0.39, *p* < 0.001), and the 5-year OS was 82% against 69%, respectively (HR = 0.51, *p* = 0.01) [120].

In contrast, consolidation therapy did not prove beneficial in the phase III BMT CTN00702 STaMINA trial. A total of 758 patients were randomly assigned to receive one of 3 treatments within a year: single ASCT + lenalidomide maintenance, tandem ASCT + lenalidomide maintenance, and ASCT followed by 4 cycles of VRD as consolidation, with lenalidomide maintenance [121]. The 5-year PFS for each trial arm was 47.5%, 44.1%, and 45%, respectively, following a median follow-up of 76 months (*p* = 0.685). There were no significant differences in OS across study arms (5-year 74.7%, 75.4%, and 76.4%, respectively; *p* = 0.745) [122]. The majority of patients in the StaMINA research received VRD regimens, and induction treatment continued up to 1 year. In the EMN02/HOVON95 study, tandem ASCT significantly improved 5-year PFS (53.5% with tandem vs. 44.9% with single, hazard ratio (HR 0.74, *p* = 0.036) and 5-year OS (80.3% vs. 72.6%, respectively). In contrast, the StaMINA trial found no benefit from tandem ASCT [123]. As a result of these contradictory findings, the latest ESMO guidelines do not advise consolidation therapy including tandem ASCT as a standard course of treatment for all patients following ASCT [124].

Dhakal et al. (2018) appraised the ASCT role in MM patients, where they reviewed HDC followed by ASCT with standard-dose therapy (SDT) utilizing novel agents evaluated in phase III RCTs [125]. However, several studies were conducted prior to the development of recent induction therapy using immunomodulating medications and proteasome inhibitors (also known as novel agents). The results showed that in several studies, HDC followed by ASCT was associated with higher response rates (RR), PFS, and even OS (Attal et al., 1996) (Child et al., 2003) (Fermand et al., 2005) [126,127,128]. All recent prospective studies (Palumbo et al., 2014) (Gay et al., 2015) (Cavo et al., 2016) (Attal et al., 2017) have constantly found a PFS advantage, although their impact on RR and OS in comparison to SDT has varied [129,130,131,132].

Furthermore, some of these inconsistencies were associated with the use of inadequate induction regimens, single HDC followed by ASCT (HDC1) versus tandem HDC followed by ASCT (HDC2) in the study arms, and insufficient follow-up throughout the study. Particularly in light of recent findings from two large RCTs that yielded conflicting results, the role of HDC2 compared to HDC1 remains uncertain [133,134]. Compared to SDT, HDC followed by ASCT was better related to PFS with few side effects. Moreover, the PFS for tandem HDC followed by ASCT and single HDC followed by ASCT with bortezomib/lenalidomide/dexamethasone were both superior to those of single HDC followed by ASCT alone and SDT. However, all four strategies showed similar OS. However, an OS benefit may be more clearly defined with a lengthier follow-up period, although this will likely depend on how well recurrence therapy works [125].

Mina and Lonial (2019) conducted a review to find a role for ASCT in MM [135]. However, they compared ASCT to SDC without transplantation based on clinical trials (Attal et al., 1996) (Child et al., 2003), with the conclusion that HDC followed by ASCT significantly increased PFS and OS compared to SDC [126,127]. Following this, various studies were carried out to demonstrate the advantages of HDC followed by ASCT over SDT, but only one of these studies was able to identify a substantial advantage of OS for patients receiving ASCT [128,136,137,138,139]. None of the trials reported prior to 2010 that compared ASCT with SDT used novel agents for the initial therapies of recently diagnosed MM patients. Keeping this in view, patients randomly allocated to the ASCT group in the European EMN02/HO95 study showed a greater 3-year PFS rate (66% vs. 58%; *p* = 0.037) when contrasted with patients in the VMP arm. The trial compared ASCT with one or two treatments of HDM at 200 mg/m^2^ and consolidation with melphalan, prednisone, and bortezomib (VMP) following a bortezomib-based induction [133].

In the IFM 2009 trial, a formal comparison between RVD and ASCT was conducted. Following 3 RVD induction cycles, 700 patients were randomly allocated to receive lenalidomide maintenance or 1 course of HDM (200 mg/m^2^) followed by ASCT, and then 2 RVD cycles or 5 RVD cycles without ASCT. In the ASCT arm, there was a 35% lower risk (median PFS, 50 vs. 36 months; *p* < 0.001) for those receiving a transplant compared to those who received RVD alone. This was due to a higher rate of CR (59% vs. 48%; *p* = 0.03) and MRD negativity (79% vs. 65%; *p* < 0.001). After four years, there was no significant difference in OS. Nevertheless, an extended follow-up may be necessary to identify a difference in OS among the two groups, particularly given the abundance of salvage therapy alternatives that might obscure the OS advantage. Furthermore, earlier ASCT has increased PFS in every trial, which implies that most patients would have better control of their disease and a deeper response [132].

A phase III trial carried out in Italy showed that for six cycles following an induction therapy consisting of four cycles of lenalidomide + dexamethasone (RD) in all patients, tandem ASCT was preferable to MPR (melphalan, prednisone, and lenalidomide). Patients who underwent ASCT exhibited significantly improved PFS with a median of 43 months compared to 22.4 months for those who did not (HR = 0.44, *p* < 0.001). Additionally, the overall survival (OS) at four years was notably higher in the ASCT group at 81.6% compared to 65.3% in the non-ASCT group (HR = 0.55, *p* = 0.02) [129]. In a related phase III trial conducted by EMN, patients were randomized to receive either six cycles of cyclophosphamide, lenalidomide, and dexamethasone (CRD) or undergo ASCT after receiving four cycles of RD as induction. In this trial, ASCT demonstrated significant improvements over CRD in terms of PFS with a median of 42 months compared to 28 months (HR = 0.67, *p* = 0.014), as well as OS at 4 years, with 87% versus 71% (HR = 0.51, *p* = 0.028) [130].

However, in all the trials conducted so far comparing ASCT to novel agent therapies without transplantation for patients with newly diagnosed MM, ASCT continues to show a preference for high-quality responses and prolonged PFS. Moreover, two trials have also identified a significant advantage in OS for patients undergoing ASCT [136,137,138,139]. Hence, ASCT remains the cornerstone of treatment for individuals newly diagnosed with multiple myeloma [135].

In a review published in 2022, Nunnelee et al. conducted a retrospective survival analysis on recently diagnosed MM patients at Ohio State University who received ASCT between 1992 and 2016. A total of 1001 MM patients were eligible. Patients were divided into 5 groups according to historical advancements in novel therapies for the treatment of MM. Between 1992 and 2016, PFS (*p* < 0.01) and OS (*p* < 0.01) statistics showed considerable improvements. PFS and OS significantly improved for patients under 65 (*p* < 0.001 and *p* = 0.002) and for those above 65 (*p* < 0.001 and *p* = 0.001). Both high-risk patients (*p* = 0.019 and *p* < 0.001) and standard-risk patients (*p* < 0.001) demonstrated improvements in PFS and OS. The post-transplant response significantly improved over time (*p* < 0.01). This shows that the survival rates of MM patients have significantly increased as a result of the introduction of novel agents and post-ASCT maintenance [140].

#### 2.4.6. Role of Tandem ASCT in Treating MM

A tandem ASCT could be suggested for the treatment of MM, either soon following the initial ASCT as consolidation therapy or as salvage therapy in patients who have recurred after a prior ASCT [141]. Given the variety of regimens that have been authorized for use during the treatment of these patients, it is questionable whether a second ASCT is necessary for MM patients who recur. The most up-to-date recommendations have been released by the European as well as American Society of Blood and Marrow Transplantation, along with the Blood and Marrow Transplantation Clinical Trials Network (BMT CTN), in 2015 [142]. However, a tandem ASCT may be considered for patients who have been in remission for more than 18 months following an initial ASCT. Nevertheless, these recommendations appear to be out of date when taking into account the median PFS achieved with a strategy involving induction regimens through triplets or quadruplets, consolidation, and maintenance therapies. Similarly, the two RCTs comparing a non-transplant strategy with salvage are outdated. This issue could be addressed by prospective randomized trials comparing salvage ASCT with the most effective non-transplant strategy [143].

To summarize, the role of ASCT in MM has been well examined, and several RCTs have shown that it outperforms SDT in terms of greater RR, PFS, and even OS. The findings of these investigations have shown some inconsistencies, particularly when comparing single HDC followed by ASCT (HDC1) with tandem HDC followed by ASCT (HDC2) and when using insufficient induction regimes. Despite the variable effects on RR and OS compared to SDT, recent prospective studies consistently revealed a PFS advantage for HDC followed by ASCT.

#### 2.4.7. The Role of CAR-T Cell for the Treatment of MM

The role of ASCT will most likely be modified in the near future, with the approval of CAR-T cell therapy for MM. The FDA has approved the first anti-BCMA CAR-T cell treatment, idecabtagene vicleucel (Ide-Cel), for recurrent MM patients who have received at least three lines of prior therapy. Ongoing research is now exploring its use in newly diagnosed MM [144,145]. In a phase I study, CAR-T cells have been utilized as a consolidated approach for high-risk newly diagnosed MM patients after at least three cycles of induction treatment. The anti-BCMA CAR-T drug cletacabtagene autoleucel, also known as cleta-Cl, has not yet received approval. However, in the CARTITUDE-1 study, it demonstrated an ORR of almost 98% in a cohort of recurrent MM patients who had undergone extensive pretreatment [146]. The primary objective of the phase III CARTITUDE-5 study is to evaluate PFS in newly diagnosed MM patients for whom ASCT is not intended as the first treatment. Patients are randomized to receive VRD-RD and VRD followed by Cilta-Cel infusion. Additionally, the phase 1/2 SZ-CART-MM02 trial is investigating tandem ASCT using anti-CD19 and anti-BCMA CAR T-cell infusion in high-risk newly diagnosed MM patients [147]. In conclusion, it remains uncertain whether CAR-T cell therapy could replace the role of ASCT as a consolidation approach in newly diagnosed MM, especially for high-risk patients, given the promising results in the recurrent MM setting and the ongoing studies.

### 2.5. The Role of the ASCT in Treatment of the AL Amyloidosis

Plasma cell disorders, also referred to as plasma cell dyscrasias, encompass a diverse group of diseases characterized by the abnormal proliferation of plasma cells [148]. A systemic disease known as AL amyloidosis (amyloid light chain; formerly known as primary amyloidosis) is distinguished via the amyloid deposition method that affects several organs, and results in poor patient survival [149]. Moreover, the fibrils that deposit and aggregate in several tissues are made up of the monoclonal light chains kappa or lambda, or their fragments [150]. Despite a consistent incidence, there was an increase in the number of cases from 15.5 cases per million in 2007 to 40.5 cases per million in 2015 [151]. Although AL amyloidosis can be diagnosed at any age, its prevalence significantly increases after the age of 40 [152]. The majority of cases are identified in men, with an average age of 65 years [153]. When deciding on a course of therapy for a patient, the first thing to determine is whether or not the patient qualifies for ASCT. Therefore, since the beginning of the 1990s, the preferred therapy for low-risk patients with AL amyloidosis continues to be HDC of melphalan followed by ASCT [154,155,156].

Sanchorawala et al. (2003) carried out a prospective RCT that aimed to investigate the time of HDC followed by ASCT in AL amyloidosis. A total of 100 recently diagnosed AL amyloidosis patients were randomly placed into 2 arms. The first arm received HDC followed by ASCT as the initial therapy, while the second arm received two sessions of oral melphalan + prednisone. The trial aimed to compare clinical and hematological outcomes, as well as survival. The results indicated that the overall response was not statistically different between the two therapy arms (*p* = 0.39), following a 45-month median follow-up period. After treatment, neither group exhibited differences in hematologic response or improvements in organ systems. Fewer patients in Arm-2 received HDC followed by ASCT due to disease progression during the oral chemotherapy stage, which excluded them from subsequent HDC. This significantly impacted cardiac patients and led to an early survival disadvantage in Arm-2. Consequently, early treatment with oral melphalan + prednisone did not confer benefits to recently diagnosed AL amyloidosis patients eligible for HDC followed by ASCT. Therefore, delaying HDC followed by ASCT due to early oral chemotherapy was associated with a worse chance of survival, especially for patients with cardiac involvement [157].

Sanchorawala (2020) conducted a review that aimed to provide a comprehensive assessment of long-term outcomes in regard to survival, hematologic response, and recurrence, in addition to organ responses after ASCT. Recently, numerous centers have reported on the prolonged results of several patients having AL amyloidosis undergoing HDC followed ASCT. However, according to the Mayo Clinic, the predicted 15-year OS was 30%. However, it was much longer for patients who achieved a hematological CR at 19.3-years as opposed to 5-years among those who did not. Then, in this series, the lack of heart involvement, obtaining complete HDC, and achieving hematologic CR all continued to be independently related to survival after ASCT in AL amyloidosis. In addition, according to a new publication from the UK National Amyloidosis Center, in 264 patients who underwent HDC followed by ASCT for AL amyloidosis, the median of OS was 8.2-years, as well as 33% long-term survival at 20 years [158]. Therefore, patients receiving HDC followed by ASCT for AL amyloidosis had a better 5-year OS, according to extensive registry information from 134 centers that was gathered by the CIBMTR. The authors demonstrated that RCT with non-ASCT regimens is required because the post-ASCT survival rate has been significantly raised in the past few years while early treatment-related mortality has decreased. In contrast, only a RCT phase III was unable to demonstrate a survival advantage for HDC followed by ASCT [159]. However, in this study, a sizable part of the patients randomly assigned to the HDC followed by the ASCT arm were not really transplanted. The transplant arm experienced severe toxicity with a premature death rate of 26%, and the follow-up was limited [160].

The comprehensive review by Al Hamed et al. (2021) highlights the evolving perspectives on the selection of therapy for ASCT-eligible patients over the years. ASCT offers high response rates and durable responses, but its drawbacks include the potential for severe short-term morbidity and mortality, limiting its applicability to a small number of properly selected patients [161]. The CR rates attained with ASCT continue to be higher than those achievable using any other medication regimen, and findings from two significant studies indicate that the CR rates following ASCT were dose-dependent. The study conducted by Cibeira et al. (2011) reports on patients with and without CR after analyzing a series of 421 sequential patients that underwent HDC followed by ASCT at a single referral site. Overall TRM was 11.4% (5.6% over the previous five years). Intention-to-treat analysis revealed that the median EFS was 2.6-years and OS was 6.3-years, while the CR rate was 34%. Again, after HDC followed by ASCT, 81 individuals passed away within a year without having their hematologic and organ responses assessed. Moreover, 43% of the 340 patients reached CR, and 78% of them had an organ response. For CR patients, the median EFS was 8.3-years, and the OS was 13.2-years. The median EFS and OS for the 195 patients who did not reach CR were 2 and 5.9 years, respectively. In total, 52% of them showed an organ response. This shows the possibility for lasting hematologic and organ responses to HDC followed by ASCT in AL amyloidosis, even among those who fail to achieve CR, which results in an elevated organ response as well as a longer OS [162].

Similarly, Tandon et al. (2017) revisited the conditioning doses in recently identified AL amyloidosis receiving first line ASCT and concluded that it influences both the responses and survivals. Among 457 patients, 314 (69%) underwent full-intensity therapy (FIT), while 143 (31%) underwent reduced-intensity therapy. The data demonstrated that in comparison to reduced-dose conditioning, the full-dosage conditioning was associated with prolonged rates of excellent partial response (79% versus 62%; *p* < 0.001), CR rate (53% versus 37%; *p* = 0.003), and 4-years PFS (55% versus 31%; *p* > 0.001) and 4-years OS (86% versus 54%; *p* > 0.001) were both better in the FIT group than in the reduced-dose group. The outcomes of the multivariate analysis supported the independent impact of conditioning dosage on PFS as well as OS. The authors claim that the usage of reduced-dose conditioning should be reevaluated in light of the findings of this investigation [163]. To sum up, high-dose melphalan was linked with reduced TRM in both studies (9% versus 14%, *p* = 0.12; 2% versus 6%, *p* = 0.01). In addition, the median of the PFS and OS were both meaningfully higher in the high-dose groups [162,163].

However, considering the greater responses obtained with ASCT in comparison to other therapy methods, it is critical to evaluate the strength that comes from such response. Furthermore, a study by Muchtar et al. (2019) examined the necessity for further or second line therapy in 186 patients following ASCT. It identified that 47% of people with the disease who lived for a minimum of 10 years after diagnosis had not received any treatments. When comparing ASCT therapy against standard-intensity therapies as the first therapeutic approach, 58% of ASCT patients remained treatment-free at 10 years, as opposed to just 36% in the non-ASCT group (22%). The findings showed long-term survivors among patients with AL amyloidosis, and ASCT was associated with a more prolonged and durable treatment response compared to standard-intensity therapy [164].

Nevertheless, an RCT including 100 patients from the years 2000 and 2005 found evidence that was opposed to ASCT for AL amyloidosis. It compared HDC followed by ASCT to high-dose standard-intensity therapy with melphalan + dexamethasone. However, the non-ASCT group exhibited a greater survival rate (*p* = 0.04). Despite the longstanding use of HDC followed by ASCT, some participating sites had only recently adopted this therapeutic approach, contributing to the study’s limitations. The criteria for patient selection lacked consistency, and only 74% of the patients assigned to the HDC followed by ASCT arm ultimately underwent ASCT. This subgroup experienced a high transplant-related mortality (TRM) of 24%, and most survivors succumbed to early mortality [159]. Similar to this, Mhaskar et al. (2009) conducted a systematic review of 12 studies, which revealed no benefit of ASCT over CHT in terms of prolonging OS in AL amyloidosis patients. Due to weak evidence, the authors suggested further research [165].

The necessity of induction chemotherapy for ASCT in AL amyloidosis seems to be another contentious issue. A review (by Minnema and Schönland, 2019) summarized recent works in the last decade, indicating that induction therapy, when used more extensively before ASCT, tends to achieve superior hematologic outcomes than ASCT alone [166]. The study by Cibeira et al. (2011), as mentioned earlier, demonstrated that even for patients unable to achieve CR, treating a subset of AL patients with HDC followed by ASCT led to a significant organ response rate and prolonged OS [162].

Another study by Huang et al. (2014) aimed to compare HDC followed by ASCT alone with two cycles of bortezomib + dexamethasone (BD) followed by HDC and ASCT as induction therapy for recently diagnosed AL amyloidosis patients. In this study of 56 patients, the BD + HDC followed by ASCT arm showed a higher rate of CR at 12 and 24 months (67.9% and 70%, respectively) compared to HDC followed by ASCT alone (35.7% and 35%, respectively; *p* = 0.03). The PFS and OS varied significantly across the two groups, with higher rates observed in the BD + HDC followed by ASCT group. The 2-year OS was 95.0% in the BD + HDC followed by ASCT arm and 69.4% in the HDC followed by ASCT alone arm (*p* = 0.03). The PFS was 80.7% in the BD + HDC followed by ASCT arm and 51.1% in the HDC followed by ASCT alone arm (*p* = 0.01). In conclusion, the preliminary findings suggest that BD induction + HDC followed by ASCT produces better results than HDC followed by ASCT alone in treating AL amyloidosis, but further research is needed to determine the prolonged advantages of this treatment [167].

Parmar et al. (2014) conducted a retrospective analysis with a 14-year follow-up to address the challenge of determining the proper treatment strategy for AL amyloidosis. While ASCT has been linked to prolonging survival, the failure of randomized studies to demonstrate an overall survival (OS) benefit raises concerns about its effectiveness. The analysis compared outcomes of ASCT to CHT in AL amyloidosis patients treated at their facility. The expected 5-year OS was 63% versus 38% for ASCT versus CHT, and the expected 10-year OS was 56% versus 10%. Factors, such as age under 60, induction therapy with newer agents, kidney-alone involvement, and ASCT were associated with better OS in multivariate analysis, highlighting the correlation between long-term OS and ASCT [168]. D’Souza et al. (2015) conducted a retrospective study on 1536 AL amyloidosis patients undergoing ASCT at 134 facilities. OS was examined for three time periods, showing an improvement in 5-year OS from 55% in 1995–2000 to 61% in 2001–2006 and 77% from 2007–2012 [169]. Sanchorawala et al. (2015) presented a prospective clinical trial where 35 patients underwent induction with bortezomib, followed by HDC and ASCT. Outcomes included a very good partial response rate of 63%, with median OS and progression-free survival not attained after a median follow-up of 3 years. The 5-year OS rate was 83% [170].

Hazenberg et al. (2015) conducted a prolonged follow-up of HDC followed by ASCT after vincristine + doxorubicin + dexamethasone as induction therapy in AL amyloidosis. In the prospective phase II study with 69 patients, preliminary data in 2008 showed a 4-year OS of 62% for all patients and 78% for transplant recipients. In the extended follow-up, survival was 8 years after enrollment for all patients and 10 years following transplantation for transplant recipients. The authors suggested that less harmful, non-intensive induction therapy followed by HDC and ASCT could lead to longer survival in recently diagnosed AL amyloidosis patients [171].

Landau et al. (2017) conducted a retrospective outcomes analysis of 143 patients who underwent HDC followed by ASCT with or without consolidation. The TRM rate after 100 days was 5%, and CR rates were 24% at 3 months and 48% at one-year post-ASCT. Patients who received bortezomib consolidation had a higher CR rate of 62%. After a median follow-up of 7.7 years among survivors, the median OS was 10.4 years, and the median EFS after HDC followed by ASCT was 4.04 years. The EFS (*p* = 0.01) and OS (*p* = 0.04) were considerably longer in patients having CR at 12 months following ASCT, and melphalan dosage had no effect on EFS (*p* = 0.26) nor OS (*p* = 0.11). In conclusion, HDC followed by ASCT was considered safe for a subset of patients, correlating with increased long-term survival, and remains a crucial cornerstone approach for AL amyloidosis, even with the availability of novel medications for consolidation [172].

Minnema et al. (2019) conducted a prospective, multicenter, phase II study examining the use of four sessions of bortezomib + dexamethasone as induction therapy + HDC followed by ASCT in recently diagnosed AL amyloidosis patients, with 72% having two or more affected organs. The total hematological response rate after induction therapy was 80%, with 20% complete remissions and 38% very good PR (VGPR). However, 15 patients did not qualify for ASCT due to various reasons, including side effects from the therapy and organ damage from the disease. Two of the patients died. Due to impaired renal function, four patients received a smaller dose of melphalan 200 mg/m^2^ compared to the other 31 patients. The transplantation process did not result in any deaths. Hematologic responses increased to 86% with 46% CR and 26% VGPR at 6 months post-ASCT. The study did not achieve its primary endpoint due to a high withdrawal rate of therapy before transplantation, and the intention-to-treat analysis showed a CR rate of 32%. Organ responses continued to improve after ASCT. The use of bortezomib + dexamethasone in patients with AL amyloidosis was highly efficient, but 30% of patients were unable to undergo ASCT following induction therapy due to treatment-induced toxicity and disease features [166].

In summary, ASCT remains a viable therapy option for AL amyloidosis patients, showing improved long-term survival and organ response rates. High-dose melphalan has been associated with reduced treatment-related mortality and improved outcomes. The timing of HDC followed by ASCT has been evaluated in randomized controlled trials, suggesting that delaying it with early oral chemotherapy may negatively impact survival. The role of induction chemotherapy for ASCT remains debated, with some studies showing better results with induction therapy, while recent studies have demonstrated the efficacy of using bortezomib and dexamethasone as induction therapy, followed by HDC and ASCT. Further research is needed to optimize the benefits of ASCT in AL amyloidosis patients.

## 3. Future Expectations for ASCT

ASCT has demonstrated positive outcomes in cancer treatment and there is still room for improvement. Moreover, gene-editing techniques to modify SCs before they are re-infused into the patient’s body is a promising area of research. Another encouraging avenue of investigation is the use of ASCT in combination with other therapies, such as immunotherapy, to enhance patient outcomes and reduce treatment-related toxicities [67,173,174,175,176]. Identifying biomarkers that can predict which patients will benefit from ASCT is also a promising area of research [177,178]. The patient selection process for ASCT is expected to improve with better risk classification in the coming years. This will be accomplished by improvements in molecular and genetic profiling, enabling more individualized therapy options [179]. It may involve adjusting ASCT procedures to the needs of particular patient populations or implementing tailored therapy based on particular illness characteristics [180]. Apart from this, the clinical trials in progress evaluating tandem ASCT, especially those involving high-risk patients, have indicated a survival benefit and deeper responses with tandem ASCT vs. single ASCT [176,181]. The future research will play a crucial role in elucidating the long-term outcomes of ASCT and its comparative effectiveness with other treatment modalities. The continuous progress in research and advancements indicates a promising future for this SC therapy.

## 4. Conclusions

ASCT is a type of SC transplantation that has shown varying degrees of success in treating different types of lymphomas, including MM and Al amyloidosis. According to our review of the available data for cancers, like HL, the impact of ASCT on OS is unclear. However, recent studies suggest it can be effective in treating pre-transplant chemotherapy-responsive cases. ASCT may also improve outcomes in PTCL and DLBCL, depending on patient factors. For FL, it is not recommended as first-line consolidation but can improve outcomes in recurrent cases. In MM, ASCT is superior to SDT, although recent studies have shown variable effects. In conclusion, ASCT stands as a viable option for AL amyloidosis, showing promise for enhanced survival and organ response rates. While it appears superior to alternative therapies, further research is essential to fully comprehend ASCT’s role in treating these disorders. The focus of future studies should be on expanding our understanding of the mechanisms underlying ASCT and identifying ways to enhance its efficacy and safety. Additionally, the future holds promise for extending the application of ASCT to other non-malignant conditions, along with the development of tailored protocols for solid tumors.

## Figures and Tables

**Figure 1 cells-12-02855-f001:**
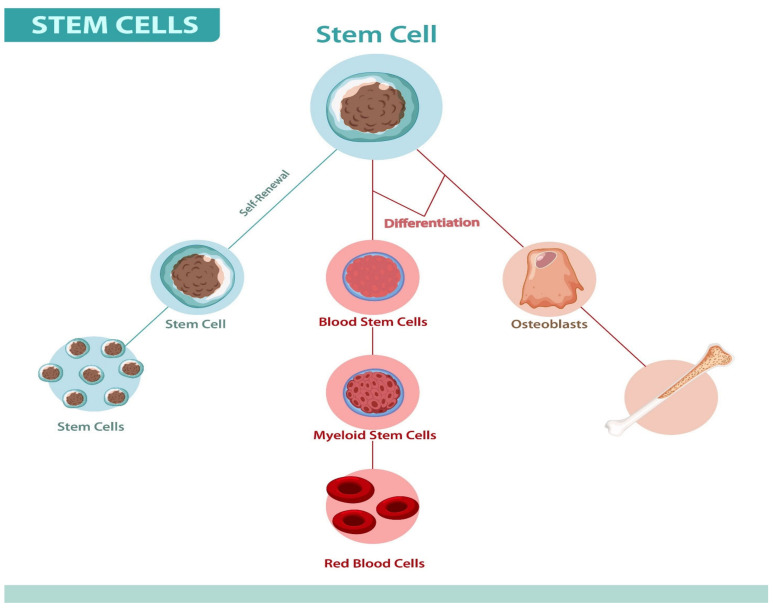
The characteristics of SCs (self-renewal and differentiation).

**Table 1 cells-12-02855-t001:** SC classification based on the potency.

Types of SCs	Definition
Totipotent	Having the capability to differentiate into every type of cell is possible. Examples include the zygote formed during egg fertilization and the initial few cells produced when the zygote undergoes division [6].
Pluripotent	Having the capacity to differentiate into practically every type of cell. Examples include embryonic stem cells (ESCs) and cells originating from the germ layers—ectoderm, mesoderm, and endoderm—during the early stages of ESC differentiation [7].
Multipotent	The capacity to develop and give rise to a closely related family of similar cells. Examples include mesenchymal stem cells (MSCs) or hematopoietic stem cells (HSCs), which can differentiate to form various blood cells, including red, white, or platelet-producing cells [8].
Oligopotent	Only being capable of differentiation into a few distinct cell types. Examples include a myeloid SC, capable of dividing into white blood cells yet not into red blood cells [9]
Unipotent	Being capable of producing just the same types of cells while possessing the necessary capacity for self-renewal to be categorized as SCs, such as dermatophytes [10].

**Table 2 cells-12-02855-t002:** SC classification based on their sources.

Types of SCs	Definition
Embryonic SCs	Self-replicating, pluripotent cells originating from embryos in the early stages of development, typically before uterine implantation. These cells originate from blastocysts, which are hollow microscopic balls of cells, and human ESCs are typically obtained from embryos aged 3 to 5 days, containing approximately 150 ESCs [14,15].
Adult SCs	Somatic or adult SCs are totipotent or multipotent, undifferentiated cells distributed through the body after embryonic development. They undergo cell division to replace dead cells and restore damaged tissues. Nevertheless, the main functions of adult SCs within an organism are the maintenance and restoration of the tissues in which they are present [16]. Initially, it was believed that tissues producing adult SCs could only generate cells of the same type; for instance, the bone marrow was thought to exclusively produce red blood cells [17].
Pluripotent SCs	SCs with characteristics resembling those of ESCs were created through the reprogramming or conversion of somatic cells to a pluripotent state. These cells are known as induced pluripotent stem cells (iPSCs) and were developed by scientists through the alteration of specific gene expression [18,19].

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
