# Peer review of "Autologous Stem Cell Transplant in Hodgkin’s and Non-Hodgkin’s Lymphoma, Multiple Myeloma, and AL Amyloidosis"

_cells, 2023, doi:10.3390/cells12242855_

Round 1

Reviewer 1 Report

Comments and Suggestions for Authors

The authors review the use of autologous stem cell transplantation in the treatment of Hodgkin lymphoma, diffuse large B cell lymphoma, follicular lymphoma, multiple myeloma and amyloidosis. The first 200 lines are too much basic, history and general schoolbook knowledge and could/should be removed.

For Hodgkin and non-Hodgkin it is mostly used as salvage therapy, in MM it is an important part of first  line therapy. The most interesting part, and thus the focus should be on MM and amyloidosis, with a sideline on the salvage therapy. The use of the word revolutionary in the title might attract readers but will dissappoint most the way the review is now.

1.       Please remove the first 200 lines

2.       Focus the review on MM and amyloidosis

3.       Review the title: it should probably be autologous stem cell transplantation, and the use of revolutionary is questionable.

4.       The focus on OS is a bit high, personally I think PFS is more important.

Comments on the Quality of English Language

The title and some minor mistakes

Author Response

  1. Please remove the first 200 lines

Author response: Done.

  1. Focus the review on MM and amyloidosis

Author response: Done.

We added more information for MM as you can see below in comment 3 of the second reviewer.

  1. Review the title: it should probably be autologous stem cell transplantation, and the use of revolutionary is questionable.

Author response: Done. It is now edited to ‘Autologous Stem Cell Transplant in Hodgkin's & Non-Hodgkin's Lymphoma, Multiple Myeloma, and AL Amyloidosis’

  1. The focus on OS is a bit high, personally, I think PFS is more important.

Author response: Yes, we agree with that. However, the articles published in recent years and included in our review mostly focus on OS. We have tried to extract information regarding PFS wherever possible and added text. Several studies mentioning values for PFS for all four types of cancers covered in this review are now mentioned.

Reviewer 2 Report

Comments and Suggestions for Authors

The review entitled: “The role of autologous stem cells transplantations: a revolutionary treatment for Hodgkin's and Non-Hodgkin's Lymphoma, Multiple Myeloma, and AL Amyloidosis” (cells-2741963) by Alnasser et al. aims to review the effectiveness and safety of ASCT in lymphomas, multiple myeloma and AL amyloidosis in the context of the current literature.

Albeit the manuscript is well written, prepared and of special interest, comments should be addressed.

Comments:

1.    Title: the title should be adapted, since the review is in parts rather a historical overview. The title suggests a new treatment.

2.    Introduction: this section should be shortened and some parts of it could be included in sections of the manuscript where appropriate.

3.    Role of ASCT in MM patients: the authors should highlight the development how patient were in former times allocated to ASCT compared to the current practice. The evaluation of a rather biological age than numerical age is an important approach (e.g. according comorbidities). The authors should add this information additionally.

4.    Role of ASCT in MM: please highlight more intensively how novel agents prior and post ASCT improved the outcome of MM patients. Moreover, the approach of CAR-T cell in the this context should be more discussed.

5.    Please provide for each table an abbreviation section below the tables. Figure 2 could be deleted.

6.    Acknowledgments: the authors should give information about “author contributions” and clarify the contribution of the “Deanship of Scientific Research”.

Author Response

  1. Title: the title should be adapted, since the review is in parts rather a historical overview. The title suggests a new treatment.

Author response: We have revised the title to clarify this. The new title is ‘Autologous Stem Cell Transplant in Hodgkin's & Non-Hodgkin's Lymphoma, Multiple Myeloma, and AL Amyloidosis’

  1. Introduction: this section should be shortened and some parts of it could be included in sections of the manuscript where appropriate.

Author response: We deleted the first 200 lines from the introduction. We have edited the whole paper and the editing of major portions can be seen in red.

  1. Role of ASCT in MM patients: the authors should highlight the development how patient were in former times allocated to ASCT compared to the current practice. The evaluation of a rather biological age than numerical age is an important approach (e.g. according comorbidities). The authors should add this information additionally.

Author response: Added.

It is reproduced below:

The number of patients undergoing autologous stem cell transplant (ASCT) in older age groups has significantly increased in recent years. While many trials historically used an age threshold of 65 years to select suitable patients for ASCT in MM, numerous studies have shown that age alone has minimal impact on the outcomes of ASCT in MM patients. For instance, an analysis by the Center for International Blood and Marrow Transplant Research (CIBMTR) involving 946 MM patients aged 70 or older at ASCT revealed that those older than 70 who underwent ASCT experienced similar antimyeloma benefits without increased non-relapse mortality, relapse rate, or progression-free survival. Another investigation involving 207 MM patients aged 70–76 who underwent ASCT at the Mayo Clinic demonstrated that ASCT was well-tolerated in this age group and exhibited non-inferior progression-free survival and overall survival compared to patients under the age of 70  [82]. Moreover, transplant suitability is determined in part by assessing severe comorbidities and fragility, both of which need cautious patient selection. Patients with conditions such as severe heart failure or poor performance status are generally not recommended for ASCT. However, it is important to note that renal impairment does not necessarily preclude the use of ASCT [83]. Additionally, a comprehensive database study indicated that patients with moderate to severe renal impairment at the time of ASCT demonstrated good tolerance for the procedure [84]. This suggests that, in some cases, renal impairment may not be a strict contraindication for ASCT, and the decision should be individualized based on the patient overall health and specific medical conditions.

  1. Role of ASCT in MM: please highlight more intensively how novel agents prior and post ASCT improved the outcome of MM patients. Moreover, the approach of CAR-T cell in the this context should be more discussed.

Prior:

The purpose of induction chemotherapy prior to HDC followed by ASCT is to reduce the burden of tumors while maintaining the highest level of tolerance and the lowest level of toxicity on healthy hematopoietic cells. This will strengthen the response rate and increase the chance of engraftment. Alkylating drugs were therefore avoided in induction and regimens based on dexamethasone, like the VAD regimen (vincristine, doxorubicin, and dexamethasone), were utilized prior to the introduction of novel agents[85].

Since novel drugs have been available, several studies have demonstrated that induction regimens with one or two novel drugs (thalidomide or bortezomib) are superior to VAD regimens in terms of elevating rates of CR, CR plus near-complete response (nCR), or VGPR both prior to and following ASCT[86-89].

Several trials have demonstrated the efficacy of combining bortezomib, dexamethasone, and thalidomide (VTD). In the GIMEMA-MMY-3006 phase III trial, patients who underwent VTD vs. TD as induction (three cycles) and consolidation (two cycles) following two phases of ASCT demonstrated superior responses compared to those who underwent TD alone (≥ PR: 93% vs. 79%, p < 0.0001; ≥ VGPR: 62% vs. 28%, p < 0.0001), which translated into a significantly improved PFS (HR = 0.62, p < 0.0001)[90].

Additionally, even though the IFM2013-04 study did not identify an important variation in CR rates, it did demonstrate the superiority of the VTD arm in terms of ORR (92.3% vs. 83.4%) as opposed to the bortezomib, cyclophosphamide, and dexamethasone (VCD) arms [91].

However, the triplet VTD didn't respond better when cyclophosphamide was added; following induction, 51% and 44% of patients in the VTD and VTDC arms, respectively, achieved nCR/CR [92].

Because of its proven value in treating high-risk patients, proteasome inhibitors like bortezomib have become indispensable in the context of VTD, which has become the standard induction regimen[93, 94].

Yet, another phase III trial by PETHEMA/GEM, contrasting VTD vs. TD vs. VBMCP/VBAD/B, proved the advantages of VTD over TD. It demonstrated enhanced results in terms of CR rates (35% vs. 14%, p = 0.001) and median PFS (56.2 months vs. 28.2 months, p = 0.01) among those receiving VTD vs. TD for six cycles as induction therapy prior to ASCT. Moreover, whereas using 3-5 cycles of VTD prior to transplant is standard procedure, using 6 cycles of VTD was linked to deeper responses. When delivering 6 cycles instead of 3–4, this must be balanced against the increased adverse effects, particularly neuropathy[95].

Likewise, the combination of lenalidomide, bortezomib, and dexamethasone [RVD] with bortezomib increased the OS (75 vs 64 months; P =.025) and median PFS (43 vs 30 months; P =.002) considerably when compared to Rd alone. Consequently, RVD is now considered the standard of therapy for those who have recently been diagnosed with MM, leading the IFM to introduce VRD as induction[95-97].

These outcomes have led to the standardizations of treatment for induction therapy in recently diagnosed MM patients who are transplant candidates to be VTD, particularly in Europe [98].

Lenalidomide was initially in contrast to Rd in the phase III SWOG S0777 study, which included participants of various ages and without the intention of early ASCT. This was done in place of thalidomide when combined with bortezomib and dexamethasone (VRD). The study's primary objective, median PFS, was found to be 41 months for VRD and 29 months for Rd following a median follow-up of 84 months (p = 0.003); OS was not met in the first arm and attained at 69 months in the second[99].

The phase III IFM 2009 trial applied three cycles of VRD as induction. On the other hand, following six cycles of VRD, subjects in the PETHEMA/GEM2012 study underwent ASCT conditioning with busulfan + melphalan as opposed to melphalan. After induction, using next-generation flow (NGF) at a level of 3 × 10−6, 83.4% of patients obtained at least PR, 66.6% at least VGPR, 33.4% CR, and 28.8% of patients achieved measurable residual disease (MRD) negative[100].

Mostly utilized in the United States, the VRD regimen has evolved into another accepted standard of therapy for patients with recently diagnosed TE MM. In a study comprising the largest patient group treated for VRD in the United States (1000 patients, 751 of whom had upfront ASCT), ORR was 97%, ≥ VGPR 67.6%, and CR 35.9% following four induction cycles. Patients who underwent ASCT had median PFS and OS of 63 months and 123.4 months, respectively, with a median follow-up of 102 months[101] .

Similar results were observed in a different "real-life" experience where patients had a median PFS of 50 months and a median OS of 101.7 months after receiving 4-6 VRD cycles with ASCT. A thorough retrospective analysis of randomized trials found that, with the same safety profile, there was a much greater > VGPR rate following six cycles of VRD (70% vs. 60%, respectively) than after six cycles of VTD. However, no prospective randomized trial has contrasted VRD with VTD[102, 103].

Although the phase III ENDURANCE study, which included patients without intending to undergo ASCT, didn’t show that carfilzomib, lenalidomide, and dexamethasone (KRD) was superior to VRD [104].

Moreover, in the FORTE phase II trial, which examined various consolidation and induction therapies according to carfilzomib either alongside or without ASCT, patients who were randomly allocated to be given four cycles of KRD as induction exhibited considerably greater ≥ VGPR, the main outcome, in comparison to those who received four cycles of carfilzomib, cyclophosphamide, and dexamethasone (KCD) (70% vs. 53%, OR = 2.14, p = 0.0002). In addition[105].

On the other hand, the advent of monoclonal antibodies (mAb) like daratumumab has once again altered the course of MM therapy, and several studies evaluated the effects of including a mAb with the aforementioned triple. In the phase III CASSIOPEIA study, the quadruplet was observed to generate at least VGPR in 65% of patients vs. 56% among those receiving VTD. The study compared four cycles of VTD vs. VTS + daratumumab (D-VTD) as induction prior to ASCT[106].

However, D-VTD and VRD weren’t contrasted directly in any randomized experiment. The phase II GRIFFIN study found that introducing daratumumab to the VRD combination (D-VRD) produced superior responses; following four cycles of D-VRD, at least VGPR was 72% and MRD negativity was 21.2%, as opposed to 56.7% and 5.8%, respectively, for VRD [107].

In addition, both the safety and effectiveness of D-VRD  vs. VRD are being compared in the phase III PERSEUS study. In the phase II MASTER trial, 90% of patients obtained VGPR following four cycles of daratumumab and KRD, with 38% showing MRD negative at a level of 10−5 and 24% at a level of 10−6 [108].

The nonrandomized MANHATTAN trial revealed a 100% ORR after eight sessions of D-KRD (weekly carfilzomib), with 71% MRD negative and 95% of patients achieving at least VGPR. This rate of excellent response with D-KRD has been shown by the study. SCs have been made available to patient candidates for ASCT following 4-6 cycles of D-KRD[109].

In the IFM 2018-01 study, at least VGPR was reached in 78% of patients with six cycles by utilizing oral ixazomib in conjunction with lenalidomide, dexamethasone, and daratumumab (IRD-Dara) as induction prior to ASCT, with an MRD negative of 28% at a level of 10−5. The daratumumab with VCD (Dara-VCD) vs. VTD combinations given as induction and consolidation following ASCT are being evaluated for effectiveness in the phase II EMN 18 investigation [110].

In summary, triplet and, more significantly, quadruplet combinations, including mAbs, can produce superior responses among patients who can receive ASCT during induction therapy, with the majority of patients reaching MRD negative at the level of 10−5. The question of whether prolonging induction treatment exceeding 4-6 cycles may result in even greater responses and eliminate ASCT entirely still has to be answered.

Post:

On the other hand, consolidation therapy has been effective in enhancing the response obtained by the use of ASCT in various recent trials. Following a median follow-up of 124 months in the phase III GIMEMA-MMY-3006 trial, patients being given VTD as induction and consolidation following two rounds of ASCT had a median PFS of 60 months, as opposed to 41 months for those given TD (p < 0.0001). At 10 years, the OS was 46% and 60%, respectively (HR = 0.68, p = 0.0068). Particularly, CR was raised from 49% to 61% after two cycles of VTD consolidation. This rise was not as significant for TD (from 40% to 47%) [111]. However, obtaining CR following ASCT has been associated with extended OS and PFS [112].

More sensitive methods for response assessment (like MRD) utilized after therapy have shown better results contrasted with conventional CR, developed as a result of the greater frequency and strength of response achieved via novel regimens [113]. Furthermore, whether MRD was detected by next-generation sequencing (NGS) or NGF, negative MRD has been linked with a survival advantage. A minimum sensitivity of 10−5 is necessary and technique sensitivity limits can affect both PFS and OS. For PFS benefits, the HRs are 0.31 at a level of 10−5 and 0.22 at a level of 10−6 [114].

Additionally, insignificant variation has been reported in the 3-year PFS among patients with standard vs. high risk cytogenetics in the PETHEMA/GEM2012 study. Patients who were MRD negative (45%) with a median detection limit of 3 × 10−6, following two VRD cycles as consolidation had an 87% PFS in comparison to 50% for those with persistent MRD (HR 0.21, p < 0.001) [115].

Multiple research studies have shown outstanding results when triplet KRD is utilized as consolidation. The sCR following eight cycles of KRD (four as induction treatment and four as consolidation post ASCT) was 60% in the phase II trial from the Multiple Myeloma Research Consortium (MMRC), which included 76 patients. Of these patients, 52% obtained MRD negative. The median PFS and OS at five years, following a median follow-up of 56 months, were not achieved in patients who did not achieve MRD negativity. The parameters were 85% for PFS and 91% for OS [116]. Similarly, a phase II trial performed by IFM showed an MRD negativity of 92.6% and 63% at 2.5 x 10−5 and 10−6 levels, respectively. This was alongside sCR rate of 62% following consolidation with four KRD cycles. Five-year PFS was 45.1% in all groups, with a median follow-up similar to the MMRC trial (60.5 months), having approximately 60% MRD-negative patients and 35% MRD-positive patients [117].

In the CASSIOPEIA and GRYPHON investigations, quadruplet regimens (containing daratumumab, such as D-VTD and D-VRD (two cycles)) delivered following ASCT, have depicted enhanced responses. D-VTd substantially outperformed VTD induction/consolidation in terms of MRD negative rate (33.7% vs. 19.9%, p < 0.0001) and ≥ one year sustained MRD negativity rate (50.1% vs. 30.1%, p < 0.0001) in patients who achieved at least CR following consolidation [118]. In the GRYPHON study, MRD negativity for at least 1 year was considerably greater in the D-VRD arm than in the VRD arm (44.2% vs. 12.6%, p < 0.0001), depicting a 3-year PFS of 88.9% (vs. 81.2% for VRD). MRD negativity at level of 10−5 went up from 22% following induction to 50% after consolidation [119]. The impact of consolidation therapy in the application of ASCT strategy has not been extensively deliberated in prospective studies. With a total of 1,503 patients, the EMN02/HOVON95 study contrasted two cycles of VRD vs. no consolidation following escalation with either ASCT (single or tandem) or bortezomib, melphalan, or prednisone (VMP). Consolidation was linked to a substantially longer PFS (median 59.3 months vs. 42.9 months, HR = 0.81, p = 0.016) following a median follow-up of 74.8 months. The OS was not reached in either arm, despite the OS curve showing a favorable trend for consolidation after 5.6 years [120].

Remarkably, following consolidation, the median PFS for patients with MRD negative was 87 months in comparison with 38 months for MRD positive patients (HR = 0.39, p < 0.001), and the 5-year OS was 82% against 69%, respectively (HR = 0.51, p = 0.01) [121].

In contrast, consolidation therapy did not prove beneficial in the phase III BMT CTN00702 STaMINA trial. 758 patients were randomly assigned to receive one of three treatments within a year: single ASCT + lenalidomide maintenance, tandem ASCT + lenalidomide maintenance, and ASCT followed by four cycles of VRD as consolidation, with lenalidomide maintenance [122]. The 5-year PFS for each trial arm was 47.5%, 44.1%, and 45% respectively, following a median follow-up of 76 months (p = 0.685). There were no significant differences in OS across study arms (5-year 74.7%, 75.4%, and 76.4%, respectively; p = 0.745) [123]. The majority of patients in the StaMINA research got VRD regimens, and induction treatment continued up to 1 year. In the EMN02/HOVON95 study, tandem ASCT significantly improved 5-year PFS (53.5% with tandem vs. 44.9% with single, hazard ratio = 0.74, p = 0.036) and 5-year OS (80.3% vs. 72.6%, respectively). In contrast, the StaMINA trial found no benefit from tandem ASCT [124]. As a result of these contradictory findings, the latest ESMO guidelines do not advise consolidation therapy including tandem ASCT as a standard course of treatment for all patients following ASCT [125].

Keeping this in view, patients randomly allocated to the ASCT group in the European EMN02/HO95 study showed a greater 3-year PFS rate (66% vs. 58%; P =.037) when contrasted with patients in the VMP arm. The trial compared ASCT with one or two treatments of HDM at 200 mg/m2 and consolidation with melphalan, prednisone, and bortezomib (VMP) following a bortezomib-based induction [126].

In the IFM 2009 trial, a formal comparison between RVD and ASCT was conducted. Following 3 RVD induction cycles, 700 patients were randomly allocated to receive lenalidomide maintenance or one course of HDM (200 mg/m2) followed by ASCT, and then two RVD cycles or five RVD cycles without ASCT. In the ASCT arm, there was a 35% lower risk (median PFS, 50 vs 36 months; P <.001) for those receiving a transplant compared to those who received RVD alone. This was due to a higher rate of CR (59% vs. 48%; P =.03) and MRD negativity (79% vs. 65%; P <.001). After four years, there was no significant difference in OS. Nevertheless, an extended follow-up may be necessary to identify a difference in OS among the two groups, particularly given the abundance of salvage therapy alternatives that might obscure the OS advantage. Furthermore, earlier ASCT has increased PFS in every trial, which implies that most patients would have better control of their disease and a deeper response [136].

A phase III trial carried out in Italy showed that for six cycles following an induction therapy consisting of four cycles of lenalidomide + dexamethasone (RD) in all patients, tandem ASCT was preferable to MPR (melphalan, prednisone, lenalidomide). Patients who underwent ASCT exhibited significantly improved PFS with a median of 43 months compared to 22.4 months for those who did not (HR = 0.44, p < 0.001). Additionally, the overall survival (OS) at four years was notably higher in the ASCT group at 81.6% compared to 65.3% in the non-ASCT group (HR = 0.55, p = 0.02) [137]. In a related phase III trial conducted by EMN, patients were randomized to receive either six cycles of cyclophosphamide, lenalidomide, and dexamethasone (CRD) or undergo ASCT after receiving four cycles of RD as induction. In this trial, ASCT demonstrated significant improvements over CRD in terms of PFS with a median of 42 months compared to 28 months (HR = 0.67, p = 0.014), as well as OS at four years with 87% versus 71% (HR = 0.51, p = 0.028) [138].

However, in all the trials conducted so far comparing ASCT to novel agent therapies without transplantation for patients with newly diagnosed MM, ASCT continues to show a preference for high-quality responses and prolonged PFS. Moreover, two trials have also identified a significant advantage in OS for patients undergoing ASCT [136-139]. Hence, ASCT remains the cornerstone of treatment for individuals newly diagnosed with multiple myeloma [140].

In a review published in 2022, Nunnelee et al. have conducted a retrospective survival analysis on recently diagnosed MM patients at Ohio State University who received ASCT between 1992 and 2016. A total of 1001 MM patients were eligible. Patients were divided into five groups according to historical advancements in novel therapies for the treatment of MM.  Between 1992 and 2016, PFS (p < 0.01) and OS (p < 0.01) statistics showed considerable improvements. PFS and OS significantly improved for patients under 65 (p < 0.001 and p = 0.002) and for those beyond 65 (p < 0.001 and p = 0.001). Both high-risk patients (p = 0.019 and p < 0.001) and standard-risk patients (p < 0.001) demonstrated improvements in PFS and OS. The post-transplant response significantly improved over time (p < 0.01). This shows that the survival rates of MM patients have significantly increased as a result of the introduction of novel agents and post-ASCT maintenance [141].

2.2.5.3. The Role of CAR-T cell for the Treatment of MM

The role of ASCT will most likely be modified in the near future, with the approval of CAR-T cell therapy for MM. The FDA has approved the first anti-BCMA CAR-T cell treatment, idecabtagene vicleucel (Ide-Cel), for recurrent MM patients who have received at least three lines of prior therapy. Ongoing research is now exploring its use in newly diagnosed MM [145, 146]. In phase I study, CAR-T cells have been utilized as a consolidated approach for high-risk newly diagnosed MM patients after at least three cycles of induction treatment. The anti-BCMA CAR-T drug cletacabtagene autoleucel, also known as cleta-Cl, has not yet received approval. However, in the CARTITUDE-1 study, it demonstrated an ORR of almost 98% in a cohort of recurrent MM patients who had undergone extensive pretreatment [147. The primary objective of the phase III CARTITUDE-5 study is to evaluate PFS in newly diagnosed MM patients for whom ASCT is not intended as the first treatment. Patients are randomized to receive VRD-RD and VRD followed by Cilta-Cel infusion. Additionally, the phase 1/2 SZ-CART-MM02 trial is investigating tandem ASCT using anti-CD19 and anti-BCMA CAR T-cell infusion in high-risk newly diagnosed MM patients [148]. In conclusion, it remains uncertain whether CAR-T cell therapy could replace the role of ASCT as a consolidation approach in newly diagnosed MM, especially for high-risk patients, given the promising results in the recurrent MM setting and the ongoing studies.

  1. Please provide for each table an abbreviation section below the tables. Figure 2 could be deleted.

Author response: We have added abbreviation detail at first use (whether in text or table). Hence, not repeated the abbreviations. Rest done as suggested.

  1. Acknowledgments: the authors should give information about “author contributions” and clarify the contribution of the “Deanship of Scientific Research”.

Author contributions :

          All authors have contributed as following:

  • Conceptualization: Sulaiman Mohammed Alnasser,
  • Approach: Sulaiman Mohammed Alnasser, Sulaiman Mohammed Almutairi, Abdulmalik Mohammed Alolayan
  • Software : Sulaiman Mohammed Alnasser, Sulaiman Mohammed Almutairi, Abdulmalik Mohammed Alolayan
  • Data Curation: Sulaiman Mohammed Alnasser, Sulaiman Mohammed Almutairi, Abdulmalik Mohammed Alolayan
  • Writers - Original Draft: Sulaiman Mohammed Alnasser,  Sulaiman Mohammed Almutairi, Abdulmalik Mohammed Alolayan
  • Writing - Review & Editing: Khalid Saad Alharbi,Ali F Almutairy
  • Visualization: Sulaiman Mohammed Alnasser
  • Supervision: Sulaiman Mohammed Alnasser
  • Undertaking administration: Sulaiman Mohammed Alnasser
  • Funding acquisition: Sulaiman Mohammed Alnasser, Khalid Saad Alharbi,Ali F Almutairy

Author response: Done.

Round 2

Reviewer 1 Report

Comments and Suggestions for Authors

The authors have revised the review substantially. Personally I think Table 1 and 2 are not necessary only the relevant type and source can be discussed.

Reviewer 2 Report

Comments and Suggestions for Authors

The review entitled: “The role of autologous stem cells transplantations: a revolutionary treatment for Hodgkin's and Non-Hodgkin's Lymphoma, Multiple Myeloma, and AL Amyloidosis” (cells-2741963) by Alnasser et al. aims to review the effectiveness and safety of ASCT in lymphomas, multiple myeloma and AL amyloidosis in the context of the current literature.

After revision of the manuscript, the authors addressed all my initial comments sufficiently.